# Can Bioactive Lipid Arachidonic Acid Prevent and Ameliorate COVID-19?

**DOI:** 10.3390/medicina56090418

**Published:** 2020-08-19

**Authors:** Undurti N. Das

**Affiliations:** 1UND Life Sciences, 2221 NW 5th St, Battle Ground, WA 98604, USA; undurti@bsrc.co.in; Tel.: +1-508-904-5376; 2BioScience Research Centre and Department of Medicine, GVP Medical College and Hospital, Visakhapatnam 530048, India

**Keywords:** COVID-19, SARS-CoV-2, arachidonic acid, lipoxin A4, inflammation, prostaglandin E2

## Abstract

It is proposed that the bioactive lipid, arachidonic acid (AA, 20:4 n-6), can inactivate severe acute respiratory syndrome(SARS-CoV-2), facilitate M1 and M2 macrophage generation, suppress inflammation, prevent vascular endothelial cell damage, and regulate inflammation resolution processes based on the timely formation of prostaglandin E2 (PGE2) and lipoxin A4 (LXA4) based on the context. Thus, AA may be useful both to prevent and manage coronavrus disease-2019(COVID-19).

## 1. Introduction

Despite the fact that millions of people worldwide are exposed to the severe acute respiratory syndrome (SARS-CoV-2) virus, only some get infected of which again few develop the manifestations of coronavrus disease-2019 (COVID-19). Among those who develop the clinical disease, only some develop serious disease that calls for hospitalization, oxygen therapy, and ventilatory intervention. Thus, one has to distinguish between and understand why only few are seriously affected. Similarly, children are less likely to develop serious SARS-CoV-2 infection, and those who develop serious disease suffer from multisystem inflammatory syndrome [1]. Current understanding of COVID-19 suggests that the virus uses angiotensin converting enzyme-2 (ACE-2) receptor and the cellular protease transmembrane protease serine 2 (TMPRSS2) to enter target cells. ACE-2 receptor is present in many tissues, and the highest expression was reported in the small intestine, testis, kidneys, heart, thyroid, and adipose tissue and lowest in the spleen, bone marrow, brain, blood vessels, and muscle. ACE-2 showed medium expression levels in the lungs, colon, liver, bladder, and adrenal gland. There was no difference in ACE2 expression between males and females or between younger and older persons in many tissues [2]. In tune with the distribution of ACE2 receptors and protein expression, SARS-CoV-2 was detected by immunohistochemistry and electron microscopy in airways, pneumocytes, alveolar macrophages, and hilar lymph nodes but was not identified in other extrapulmonary tissues in autopsies of fatal COVID-19 patients [3]. Most of these fatal cases had co-morbid conditions and showed superadded bacterial and viral infections suggesting that one needs to be careful in attributing all the histopathological features to COVID-19. It is possible that SARS-CoV-2 can infect various other tissues like liver, heart, spleen, kidney, bone marrow, and lymph nodes that express ACE2 receptor. To establish the ability of SARS-CoV-2 to infect these extrapulmonary tissues, more autopsy studies are needed. An ACE-2 receptor is present in several tissues. Hence, SARS-CoV-2 is able to infect many tissues. In view of this, COVID-19 clinical presentation is highly variable and at times is atypical, silent, and bewildering. Furthermore, the COVID-19 clinical picture is modified by the immune response of the affected individual. The extensive distribution of tissues that can be infected by SARS-CoV-2 and a high number of pro-inflammatory cytokines released as a result can ultimately result in systemic inflammatory response syndrome (SIRS), and this, in turn, leads to accelerated cell death in the lungs, liver, heart, kidneys, and the adrenal parenchymal organs causing multiple organ dysfunction syndrome (MODS) similar to that seen in SARS [4,5,6,7].

## 2. Coagulation Abnormalities in COVID-19

The inflammatory damage of the microvascular system (especially endothelial cells) can initiate hypercoagulability and result in abnormal activation of the coagulation system leading to the development of generalized small vessel vasculitis and extensive microthrombosis [7]. In view of this, assessing coagulation indices in those who are acutely ill (if possible, all those who have been diagnosed to have COVID-19) is recommended. It was reported that activated partial thromboplastin time (APTT), thrombin time (TT), fibrinogen levels, evaluation of fibrinolytic system in the form of measuring D-dimers and fibrin degradation products (FDP), and platelet count need to be measured to assess the thromboembolic state in the patient. Both sepsis and critically ill COVID-19 patients have elevated levels of D-dimer, prolonged PT and APTT, ARDS (acute respiratory distress syndrome), and disseminated intravascular coagulation (DIC) that is associated with increased mortality [8,9,10,11,12]. Hence, it is important to assess the status of the coagulation system in those who are critically ill due to COVID-19 by measuring (i) viscoelastic indices that include thromboelastograph (TEG); (ii) a coagulation and platelet function such as R-time for coagulation factor activity; (iii) α angle and k-time for fibrinogen function; (iv) maximal amplitude (MA) for platelet function; (v) LY30% (clot lysis after 30 min) for fibrinolytic function (wherein ACT (activated clotting time) that indicates coagulation factor function, CR (clot rate) that measures fibrinogen function, and platelet function are assessed). These abnormalities are common in COVID-19. These measurements are believed to accurately assess the overall coagulation status of the patient and show a good correlation with the assessment of coagulation factors, fibrinogen, and platelet function in critically ill COVID-19 patients [13,14,15]. In these patients (critically ill COVID-19), the following measures may be needed: appropriate anti-coagulation intervention employing an unfractionated heparin/low-molecular weight heparins; CRRT (continuous renal replacement therapy); external membrane oxygenation therapy; thrombocytopenia correction by the anticoagulant argatroban/bivalirudin, fresh frozen plasma infusion at 15 to 30 mL/kg, prothrombin complex concentrate, cryoprecipitated fibrinogen (10 mL/kg), or human fibrinogen (30–50 mg/kg) infusion; platelet transfusion if the platelet count is <20 × 109/L; recombinant factor VII; plasma exchange; artificial liver support system (ALSS) for associated liver failure, and controlled infusion of crystalloids and synthetic colloid while maintaining adequate tissue perfusion, as the situation demands [13].

## 3. AA Regulates Inflammation, Inactivates Enveloped Viruses, and Is Needed for Wound Healing

Based on our current understanding of the genome of the SARS-CoV-2 and its antigenic components, efforts are being made to develop specific anti-viral drugs, vaccines, and antibody therapies (including plasma therapy). All these efforts take time to come to the clinic. In this context, it is noteworthy that little attention has been paid to the potential role of bioactive lipids (BALs) in COVID-19. Previously, I proposed that bioactive lipids, such as arachidonic acid (AA), eicosapentaenoic acid (EPA), and docosahexaenoic acid (DHA), and their metabolites, such as pro-inflammatory prostaglandins (PGs), leukotrienes (LTs), anti-inflammatory lipoxin A4 (LXA4, derived from AA), resolvins (derived from EPA and DHA), protectins, and maresins (derived from DHA) may have a significant role in COVID-19 [16,17,18,19]. This proposal is based on the observations that AA and other BALs (i) can inactivate enveloped viruses, such as HCV, HBV, influenza, and other corona species; (ii) have the ability to facilitate the generation of M1 and M2 macrophages (M1 by pro-inflammatory BALs, such as PGs and LTs, and M2 by anti-inflammatory BALs, such as LXA4, resolvins, protectins, and maresins); (iii) suppress the production of IL-6, TNF-α and other pro-inflammatory cytokines especially by AA/EPA/DHA, PGE2, and LXA4 and enhance the formation of IL-10 by LXA4; (iv) AA/EPA/DHA/PGE1/LXA4 have cytoprotective actions and, thus, protect normal cells from both endogenous and exogenous toxins including viruses; (v) AA/EPA/DHA/LXA4/resolvins, protectins, and maresins have vasodilator, platelet anti-aggregator actions and suppress leukocyte activation, adherence, and their ability to release free radicals, and finally (vi) BALs especially LXA4, resolvins, protectins, and maresins resolve inflammation and enhance wound healing [20,21,22,23]. Thus, availability, formation, and release of appropriate amounts of AA/EPA/DHA (especially AA) and maintenance of a balance between pro- and anti-inflammatory BALs is expected to be of benefit in the prevention and amelioration of COVID-19. Of all the BALs, AA seems to be of significant benefit in this context, since it forms the precursor to both pro-inflammatory PGE2 and anti-inflammatory LXA4. The balance between PGE2 and LXA4 is needed to maintain normal homeostasis so that inappropriate inflammatory events do not occur due to an excess of PGE2 and/or deficiency of LXA4 that could lead to severe illness. Hence, factors that regulate the formation of PGE2 and LXA4, such as desaturases, PLA2 activity that is needed for its (AA) release from the cell membrane lipid pool, activities of cyclo-oxygenase-2 (COX-2) and lipoxygenases (LOXs) that are essential for the conversion of AA to PGE2 and LXA4, and their degradation enzymes, such as 15-PGDH, have a critical role in COVID-19 [17,18]. Furthermore, AA has cytoprotective actions [24,25] (which especially protects vascular endothelial and neuronal cells), suppresses platelet and leukocyte activation, and thus, prevents thromboembolic complications seen in COVID-19.This proposal implies that a deficiency of AA (and EPA and DHA; AA > EPA ≥ DHA) may enhance the susceptibility of an individual to SARS-CoV-2 infection and administration of these fatty acids may enhance the recovery process. This is supported by the reports that human cells infected with coronavirus release large amounts of AA and LA, and these fatty acids inactivate the virus [26,27]. Previously, we showed that patients with type 1 and type 2 diabetes mellitus, hypertension, coronary heart disease, and insulin resistance have AA deficiency [28,29,30,31,32,33,34]. This accounts for the high degree of mortality seen in these patients when they develop COVID-19.

## 4. Conclusions and Therapeutic Implications

In the light of the preceding discussion, I suggest that the administration of AA in the prevention and management of COVID-19 need to be considered seriously. It is likely that the lower the plasma and tissue levels of AA are, the higher the severity of COVID-19. AA is non-toxic, can be given orally and intravenously, and its administration enhances the formation of LXA4, an anti-inflammatory molecule that also has anti-viral and anti-bacterial actions similar to AA [16,17,18,19,35].

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
