# Peer review of "Can Bioactive Lipid Arachidonic Acid Prevent and Ameliorate COVID-19?"

_1010-660X, 2020, doi:10.3390/medicina56090418_

Round 1

Reviewer 1 Report

The topic undertaken by the authors is very current and interesting, there is an urgent need to look for both drugs, vaccines and all kinds of measures supporting treatment, but also helping to prevent COVID-19 infection.

Some minor comments are as follows:

  • references should be cited in the text using square brackets not superscripts
  • reference list should be edited according to journal requirements included in the instructions for authors
  • authors contributions, funding, acknowledments and conflict of interest sections should be completed
  • line 24 - remove "Since"
  • line 30 - remove "conducting"
  • line 35 - "...other tisues, like liver...."
  • ;one 46 - "can initiate" - what initiate?
  • line 61 "PF (platelet function) indicates platelet function" ??? The fragment should be rewritten
  • line 101 - remove the brackets
  • line 110 - "be given" instead of "begiven"

Reviewer 2 Report

This is an excellent review on a very timely topic. One minor suggestion is that the author should cite at least 3-4 original Serhan references for the original discovery of the lipoxins, resolvins, maresins and protectins.

Reviewer 3 Report

This manuscript is poorly written and difficult to follow.  It has many errors such as the following examples, thus needs a complete edit and rewrite. Currently it can only be considered a working draft.

Line 17 should not read “several subjects”; millions of people worldwide is correct.

Line 24 Does not make sense starting with “Since.”

Line 32 should read “autopsies” or “postmortem examinations”

Lines 38 and 39 make no sense

Line 44 should read “that” not “those”

Line 49 should read “been” not bene

Lines 53 to 55 do not make sense

Lines 64 to 72 are an extensive list that should be enumerated or bulleted, and there should be colon after “needed”.

Line 83 enumeration (i) needed to come after “AA and other BALs” in order for the enumerated sentence to make sense.

The following lines sarehard to follow and makes little sense.  If these factors have a “a critical role in COVID-19” then this statement MUST be referenced!

97 and anti-inflammatory LXA4. Thus, factors that needed for its formation such as desaturases, PLA2

 98 activity that is needed for its (AA) release from the cell membrane lipid pool, activities of cyclo

99 oxygenase-2 (COX-2) and lipoxygenases (LOXs) needed for the conversion of AA to PGE2 and LXA4, 100 and their degradation enzymes (such as 15-PGDH) have a critical role in COVID-19.

Most importantly, there are few in Western countries who are acutely AA deficient, unlike the case with n-3 LC-PUFA.  For this reason, at least a few examples must be given proving that Covid-19 and/or SARS post mortem patients were found to be AA deficient.  There is not even any general reference given to LC-PUFA status and viral susceptibility and morbidity.

Round 2

Reviewer 3 Report

There are  a few reference citations that need attention--missing brackets, perriod instead of comma.

Overall the author has made a good faith effort to improve the work.

Author Response

The corrections suggested have been incorporated in the revised manuscript. The corrections have been highlighted in red. 

While making these corrections, some minor editing has also been done to the manuscript that has been highlighted in red. 
